# The association between routine immunisation and COVID-19 vaccination in small Island developing states

Cyra Patel[1,2]*, Gizem Bilgin[1], Andrew Hayen[3], Martyn Kirk[1], Akeem Ali[4], Aditi Dey[5,6], Ginny Sargent[1], Meru Sheel[1,2,7]*

1 National Centre for Epidemiology and Population Health, Australian National University, Canberra, Australia, 2 Sydney School of Public Health, Faculty of Medicine and Health, The University of Sydney, Camperdown, New South Wales, Australia, 3 School of Public Health, University of Technology Sydney, Ultimo, New South Wales, Australia, 4 World Health Organization, North Macedonia, 5 National Centre for Immunisation Research and Surveillance, Sydney Children's Hospitals Network, Westmead, New South Wales, Australia, 6 Sydney Medical School, Faculty of Medicine and Health, The University of Sydney, Westmead, New South Wales, Australia, 7 Sydney Institute for Infectious Diseases, Faculty of Medicine and Health, The University of Sydney, Westmead, New South Wales, Australia

* Cyra.Patel@anu.edu.au (CP); Meru.Sheel@sydney.edu.au (MS)

## Abstract

### Objectives

Understanding the link between routine immunisation (RI) performance and vaccination during an epidemic can provide insights on health systems resilience and investments to strengthen health systems. We examined the relationship between RI performance and COVID-19 vaccination coverage in small island developing states (SIDS).

### Methods

We analysed immunisation and health system performance data in 55 SIDS. Our primary outcome was COVID-19 vaccination coverage at four timepoints (June 2021, December 2021, June 2022 and December 2022). We examined associations with coverage of six childhood immunisations (5-year mean annual coverage for 2015–2019), pandemic-related disruptions to RI, new vaccine introductions, health system performance measures, and economic and demographic characteristics. We calculated Spearman correlation coefficients (r) with p-values (p < 0.05 considered significant) and 95% confidence intervals for continuous variables and mean COVID-19 vaccination coverage by categorical variables.

### Findings

COVID-19 vaccination coverage was higher in countries that sustained pre-pandemic RI coverage during the pandemic, and where HPV, influenza and measles-containing

**Data availability statement:** All data were obtained from publicly available datasets without limitations. All analysed findings have been presented in the paper and supplementary files. Data sources for all variables are listed in Table 1 of the manuscript. Data on routine immunisation coverage, new vaccine introduction and immunisation system performance were obtained from the WHO/UNICEF Estimates of National Immunisation Coverage (WUENIC) immunisation data portal (https://immunizationdata.who.int). COVID-19 vaccination coverage data and information on COVAX status were obtained from the COVID-19 Vaccination Information Hub (https://infohub.crd.co). Data on health system performance and country demographic characteristics were obtained from the World Bank Indicators dataset (https://databank.worldbank.org/source/world-development-indicators). Data on birth registration completeness were obtained from the United Nations Children's Fund (UNICEF) (https://data.unicef.org/topic/child-protection/birth-registration/). Information on country eligibility for Gavi funding was obtained from Our World in Data (https://data.unicef.org/topic/child-protection/birth-registration/) and are also available from Gavi (https://www.gavi.org/types-support/sustainability/eligibility). Data on income classifications were obtained from the World Bank (https://datahelpdesk.worldbank.org/knowledgebase/articles/906519-world-bank-country-and-lending-groups). A list of countries classified as least developed countries was obtained from the United Nations Office of the High Representative for the Least Developed Countries, Landlocked Developing Countries and Small Island Developing States (https://www.un.org/ohrlls/content/list-ldcs). The data used in analyses were the latest available at the time of extraction (April 2024).

**Funding:** The author(s) received no specific funding for this work.

**Competing interests:** The authors have declared that no competing interests exist.

(second dose) vaccines had been introduced. There were weak correlations ($|r| < 0.4$) between coverage of COVID-19 vaccination and RI, with a few exceptions of moderate correlations with the birth dose of hepatitis B vaccine (June 2022: $r = 0.421$, $p = 0.007$; December 2022: $r = 0.438$, $p = 0.005$) and first dose of measles vaccine (December 2021: $r = 0.420$, $p = 0.002$). COVID-19 vaccination coverage was strongly correlated with the density of physicians (June 2021: 0.897, $p < 0.001$; December 2021: 0.785, $p < 0.001$) and moderately correlated with that of nurses and midwives (June 2021: 0.630, $p = 0.001$; December 2021: 0.605, $p = 0.002$). COVID-19 vaccination coverage was lower in SIDS with lower country income and development status.

## Conclusions

Countries that achieved high COVID-19 vaccination coverage also sustained RI coverage during the pandemic, demonstrating health system resilience. Our findings highlight the importance of having sufficient skilled health professionals and experience in introducing new vaccines targeting different age groups into national programs, particularly in small island settings.

## Introduction

The Immunization Agenda 2030 (IA2030), the global vision and strategy for immunisation for the decade 2021–2030, emphasises the need to strengthen health systems to attain universal health coverage of all health services including immunisation [1]. Countries achieve and sustain high, equitable routine immunisation (RI) coverage when the various components of immunisation systems align with fundamental health system components and work together effectively [2,3]. The recent advocacy and increasing investments in immunisation systems from disease-focused programs reflect this shift [1,4]. System-based approaches to achieving high, equitable RI coverage can have benefits like reaching underserved and vulnerable populations who previously had limited, if any, contact with the primary healthcare system [5,6]. However, there is limited evidence on the relationship between immunisation systems and the ability to deliver vaccines during public health emergencies caused by infectious disease outbreaks.

The COVID-19 pandemic provides an opportunity to apply a health systems lens to examine the link between RI performance and emergency vaccination. COVID-19, caused by the severe acute respiratory syndrome coronavirus 2 (SARS-CoV-2), can cause severe respiratory disease requiring hospitalisation and critical care such as ventilation, and can be fatal [7]. Its high transmissibility threatened to quickly overwhelmed health systems in the first wave of infection, leading to intense efforts to develop and deploy an effective vaccine. A recent analysis of COVID-19 vaccination coverage and immunisation program maturity found that COVID-19 vaccination coverage was 14–16% higher in countries with an adult seasonal influenza vaccination program [8]. However the study did not examine factors such as ability to maintain routine immunisation coverage while concurrently implementing COVID-19

vaccination programs (a measure of system resilience), universal health coverage index and infant mortality. These indicators measure different aspects of health and immunisation system performance and including them in such analyses provides additional insights into how strengthening routine systems can improve outcomes during an emergency. Furthermore, there was a high degree of heterogeneity in the health systems and political, socioeconomic and demographic contexts across countries, and in how COVID-19 vaccination programs were implemented, which is difficult to account for in a global analysis. In this study, we build on the existing literature by examining immunisation system factors associated with COVID-19 vaccination coverage within small island developing states (SIDS).

SIDS are a group of 57 small, remote low-lying island nations, including 29 Caribbean nations, 19 Pacific Island Countries and Territories, eight Atlantic, Indian Ocean and South China Sea (AIS) member states, and one country in southeast Asia (Timor-Leste) [9]. SIDS vary in their income level classifications from low through to high income, but are all considered to be vulnerable to systemic shocks due to social, economic, environmental and infrastructural challenges and limited resources [9]. They are characterised by small population sizes and limited health infrastructure, especially highly constrained health workforces [10–12]. For example, in the Pacific region, Vanuatu, with a population of approximately 330,000, has 48 doctors and 353 nurses (0.16 and 1.16 per 1,000, respectively), while Niue has 3 doctors and 20 nurses (1.65 and 10.60 per 1,000, respectively) for their population of just under 2,000 people [13]. In the Caribbean, Dominica has 79 doctors and 461 nurses (1.12 and 6.51 per 1,000, respectively) for almost 73,000 people [13]. Their remoteness and economic and environmental vulnerability limit their capacity to respond rapidly to large infectious disease outbreaks. The fragile nature of health systems, increased vulnerability to outbreaks and costly emergency response has been recognised by Gavi, The Vaccine Alliance in their most recent strategy for 2026–2030 [14].

Regarding responses to the COVID-19 pandemic, many SIDS relied on border closures to control the importation of COVID-19 [15–17]. Their small population sizes diminish their purchasing power with vaccine manufacturers, a significant disadvantage during the COVID-19 pandemic when supplies were limited and wealthy countries were buying up available stocks [18,19]. Additionally, they face higher freight costs to transport vaccines due to their remote geography. Vaccine distribution to the point-of-service is complicated by the need to maintain cold chain in warm climates and transport vaccines by air and boat especially to remote outer lying islands. Health system organisation and performance vary across SIDS with different financing, political and socioeconomic influences. This was evident in responses to and outcomes of the COVID-19 pandemic, with some SIDS keeping the virus at bay until late 2021 and even 2022, while others experienced high case numbers and fatalities earlier on [20–22]. Policies regarding border closures, quarantine protocols and vaccine mandates varied, as did decision-making and emergency response coordination structures [15,20]. For example in the Pacific, country-level responses were complemented by the WHO's wider Pacific Joint Incident Management Team partner coordination meetings [20]. In the Caribbean, the Caribbean Public Health Agency Caribbean Regulatory System collaborated with the WHO and Pan American Health Organization to expedite regulatory approvals of COVID-19 vaccines [23]. Acceptance of COVID-19 vaccination was also a challenge in some SIDS, with surveys in Papua New Guinea, Haiti and Dominican Republic indicating only 27.6%, 43.2% and 68.8% of individuals, respectively, intended to receive a COVID-19 vaccine [24,25]. Another survey of healthcare workers found broad (>90%) acceptance of vaccines in general, but hesitancy to receive COVID-19 vaccines among 23% of respondents [26].

There is limited data on factors associated with COVID-19 outcomes, including vaccination coverage, in SIDS. Examining immunisation system factors can provide insights into investments and strategies to strengthen immunisation systems and emergency vaccination. In this study, we examined the relationship between RI systems and COVID-19 vaccination coverage in SIDS.

## Methods

An immunisation system encompasses all components of a health system necessary to deliver vaccines, including all the organisations, institutions, resources, processes and activities involved in the delivery of immunisation programs [27]. In

this study, we focused on the systems for routine immunisation and emergency vaccination, including the system's capacity to deliver both concurrently, i.e., resilience. Key terms used in this study are defined in Box 1.

---

### Box 1.–Key definitions and assumptions.

- **Routine immunisation** comprises all vaccinations provided on a regular basis according to a country's national vaccination schedule, including those delivered via mass vaccination campaigns (i.e., supplementary immunisation activities) to close gaps in coverage. Immunisation performance is most commonly measured by vaccination coverage, with higher coverage indicative of stronger performance.

- **Emergency vaccination** occurs to mitigate an infectious disease outbreak, using a novel vaccine antigen or a routine vaccine that targets different age and risk groups. This study focuses on COVID-19 vaccination during the acute phase of the COVID-19 pandemic, i.e., between January 2020 and May 2023 when COVID-19 was considered a public health emergency of international concern [28].

- **Health system resilience** is the capacity of the health system to prepare for and effectively respond to a crisis while maintaining core functions [29]. Resilient vaccination systems, therefore, are those that can provide emergency vaccination services in response to an infectious disease epidemic, with minimal disruptions to routine immunisation services.

- **New vaccine introduction** refers to the integration of newer and underutilised vaccines into a country's national immunisation program. In this study, we focused on the second dose of measles containing vaccine (MCV2), pneumococcal conjugate vaccine (PCV), rotavirus vaccine and HPV vaccine as WHO recommends them for all countries, as well as influenza vaccine to examine the effect of integrating a vaccine targeting adulthood into national programs.

---

### Inclusion criteria

We included all SIDS (N = 57) in line with the United Nations Office of the High Representative for the Least Developed Countries, Landlocked Countries and Small Island Developing States (UN-OHRLLS) classification including 39 WHO member states and 18 non-member states [9]. We included both WHO member and non-member states as they all experience social, infrastructural and economic challenges given their remote geography, small population sizes and vulnerability to climate change. We included all SIDS that had publicly reported data on COVID-19 vaccination coverage rates. We only excluded SIDS for which COVID-19 vaccination coverage was not publicly reported.

### Data sources

We extracted the following publicly available data (Table 1):

- Monthly COVID-19 vaccination coverage data for 2021 and 2022 from COVID-19 Vaccination Information Hub, collated from the WHO Joint Reporting Form (JRF) COVID-19 vaccination module [30].

- WHO/UNICEF estimates of national immunisation coverage (WUENIC) data for the years 2015–2022, collated from countries annually via the JRF process. Data were sourced from the WHO Immunization Data Portal, including data on RI coverage, new vaccine introduction and immunisation system performance [31]. Data for RI coverage were not consistently reported for non-WHO member states, and were unavailable for new vaccine introduction status and immunisation system performance measures through the WUENIC system. Indicators on dropout of routine immunisation coverage and disruptions to routine immunisation coverage during the COVID-19 pandemic were calculated based on WUENIC estimates.

**Table 1. Independent variables included in the analysis and data sources.**

| Variable group | Variable | Variable type | Categories (for categorical variables) | Data source |
|---|---|---|---|---|
| Routine immunisation performance | Immunisation coverage,* 5-year mean (2015–2019), for:<br>• BCG vaccine<br>• Hepatitis B birth dose (i.e., given within 24 hours of birth)<br>• DTP1<br>• DTP3<br>• MCV1<br>• MCV2 | Continuous & categorical | • <80%<br>• 80–<85%<br>• 85–<90%<br>• 90–<95%<br>• ≥95% | WUENIC |
| Dropout of routine immunisation coverage | Difference in the 5-year mean (2015–2019) coverage between:<br>1. BCG vaccine and MCV1<br>2. DTP1 and DTP3<br>3. DTP1 and MCV1<br>4. DTP1 and MCV2<br>5. MCV1 and MCV2 | Continuous & categorical | • <5%<br>• 5–<10%<br>• 10–<15%<br>• ≥15% | WUENIC |
| Disruption to routine immunisation coverage | Decline in coverage of any (i.e., at least 1) of three vaccines (DTP1, DTP3, MCV1) compared to coverage in 2019, by year:<br>• 2020<br>• 2021<br>• 2022 | Categorical | • Yes<br>• No | WUENIC |
| | Decline in coverage of all three vaccines (DTP1, DTP3, MCV1) compared to coverage in 2019, by year:<br>• 2020<br>• 2021<br>• 2022 | Categorical | • Yes<br>• No | WUENIC |
| New vaccine introduction | New vaccine introduction (i.e., whether countries had introduced these vaccines in their national immunisation programs) as at 2019, for:<br>• HPV vaccine<br>• Influenza vaccine<br>• MCV2<br>• Pneumococcal conjugate vaccine<br>• Rotavirus vaccine | Categorical | Analysis 1:<br>• Yes<br>• No<br>Analysis 2:<br>• Yes, ≥5 years<br>• Yes, <5 years<br>• No | WUENIC |
| Immunisation system performance | NITAG functionality, i.e., does the country have a functional NITAG as at 2019 | Categorical | Strict definition:<br>• Yes<br>• No<br>Permissive definition:<br>• Yes<br>• No | WUENIC; definitions based on Van Zandvoort et al 2019 [32] |
| | Existence of an AEFI surveillance system | Categorical | • Yes<br>• No | WUENIC |
| Health system performance | Universal Health Coverage index | Continuous | N/A | World Bank Indicators |
| | Infant mortality rate, per 1,000 live births | Continuous | N/A | World Bank Indicators |
| | Under-five mortality rate, per 1,000 live births | Continuous | N/A | World Bank Indicators |
| | Density of healthcare workers, per 10,000 population:<br>• Physicians<br>• Nurses and midwives | Continuous | N/A | World Bank Indicators |
| | Density of hospital beds, per 10,000 population | Continuous | N/A | World Bank Indicators |
| | Government expenditure on healthcare – proportion of GDP spent on healthcare | Continuous | N/A | World Bank Indicators |
| | Birth registration completeness (proportion of births registered) | Continuous | N/A | UNICEF |

*(Continued)*

**Table 1.** (Continued)

| Variable group | Variable | Variable type | Categories (for categorical variables) | Data source |
|---|---|---|---|---|
| Economic and development characteristics# | Country income level# | Categorical | • High income<br>• Upper-middle income<br>• Lower-middle income<br>• Low income | World Bank |
| | Status as a least developed country# | Categorical | • Least developed<br>• Developing | UNOHRLLS |
| | Eligibility for funding from Gavi# | Categorical | • Yes<br>• No | Gavi |
| | COVAX status# | Categorical | • Self-financing<br>• AMC<br>• No | COVID-19 Vaccination Information Hub |
| Demographic characteristics | Country population size | Continuous | N/A | World Bank Indicators |
| | Proportion of the population living in rural areas | Continuous | N/A | World Bank Indicators |
| | Population density | Continuous | N/A | World Bank Indicators |
| | Primary gender parity index | Continuous | N/A | World Bank Indicators |

AEFI: Adverse Event Following Immunisation; AMC: Advance Market Commitment; BCG: Bacille Calmette-Guérin; COVAX: COVID-19 Vaccines Global Access; DTP: Diphtheria-Tetanus-Pertussis-containing vaccine; HPV: Human Papilloma Virus; MCV: Measles-Containing Vaccine; NITAG: National Immunisation Technical Advisory Group; SIDS: Small Island Developing State(s); UNDP: United Nations Development Programme; UNICEF: United Nations Children's Fund; UNOHRLLS: United Nations Office of the High Representative for the Least Developed Countries, Landlocked Developing Countries and Small Island Developing States; WUENIC: WHO/UNICEF Estimates of National Immunisation Coverage; WHO: World Health Organization.

*These six antigens were included as they represent the six childhood vaccines that were the initial focus of the Expanded Programme on Immunization (i.e., BCG, DTP and MCV vaccines), in addition to the birth dose of hepatitis B vaccine which is a key health system indicator, i.e., access to vaccination at birth.

#The economic and development characteristic indicators included measures of a country's wealth and development status and vulnerability to shocks that use differing criteria. Country income level categories are defined by Gross National Income (GNI) per capita based on the World Bank's classifications using the World Bank Atlas Method. Least developed country status is defined based on having low GNI, an economic vulnerability index, and having low levels of human resources (based on indicators of nutrition, health, education and adult literacy). Eligibility for Gavi funding is based primarily on GNI per capita. Eligibility for a country to participate in the COVAX Advance Market Commitment were based primarily on the country's income classification (all low and lower-middle) were eligible, with some upper-middle income countries eligible based on criteria related to economic vulnerability. Self-financing countries were high and upper-middle income countries who joined the COVAX Facility who were ineligible for development assistance. Countries that did not join the COVAX facility were classified as "no" for their COVAX status.

- World Bank Indicators data for health system performance (e.g., health worker density and childhood mortality rates) and country-level demographic characteristics (e.g., country population size).

- Country income level and development status classifications as defined by the World Bank and United Nations Development Programme, respectively.

Table 1 provides further details on data sources for independent variables. All data were extracted in April 2024. As all data were aggregated at the country-level and obtained from publicly available sources, ethical approval was not required. No ethical approval was required as this study used publicly available datasets on national-level data.

## Outcomes of interest

The primary outcomes of interest were coverage of: A) the first dose of COVID-19 vaccination, and B) a complete primary course of COVID-19 vaccination. We examined coverage at four timepoints in the first two years of the global roll-out: 1)

June 2021, 2) December 2021, 3) June 2022, and 4) December 2022. We did not include coverage for COVID-19 vaccine boosters in our study as we focused on vaccination efforts aimed at increasing population-level immunity during the acute phase of the pandemic.

**Independent variables: routine immunisation**

In this study, we focused on well-established measures of the overall performance of immunisation systems at a national level, namely indicators for vaccination coverage and introducing new vaccines in national immunisation programs [33] Table 1 lists the definitions and details of independent variables. They included:

- The 5-year (2015–2019) mean annual coverage of six vaccines, i.e., Bacillus Calmette-Guérin (BCG) vaccine birth dose, hepatitis B vaccine birth dose, the first and third dose of diphtheria-tetanus-pertussis containing vaccine (DTP1 and DTP3), and the first and second dose of measles-containing vaccine (MCV1 and MCV2);

- Dropout of vaccination coverage, i.e., the proportion of children who did not receive subsequent doses of vaccination after receiving earlier doses in the series, which is often used as an indicator of the health system's ability to consistently reach populations to achieve full vaccination; [3].

- Disruption to RI during the COVID-19 pandemic (i.e., 2020, 2021 and 2022), defined as a decline in coverage of three vaccines (DTP1, DTP3 and MCV1) in comparison to 2019 (which has been used as a benchmark year in recent assessments of global immunisation coverage) [34–37] either alone or in combination;

- "New vaccine introduction" of five new and underutilised vaccines as of 2019, namely MCV2, PCV, rotavirus, HPV and influenza vaccines, as their introduction can reflect and impact the performance of health systems [38].

**Independent variables: health system and macro factors**

We included immunisation system performance measures that focus on system strengthening initiatives in recent years. These include the functionality of a national immunisation technical advisory group (NITAG) [32,39] and established national surveillance system for adverse events following immunisation (AEFI) [40,41]. In line with our hypothesis that immunisation system performance is associated with health systems as a whole, we examined the relationship with universal health coverage (UHC) index, infant and under-five mortality rates, density of healthcare workers and hospital beds, government expenditure on healthcare and birth registration completeness.

We also included economic and development indicators (i.e., country income level, status as a least developed country, eligibility for funding from Gavi The Vaccine Alliance, and COVAX status), and demographic characteristics (i.e., country population size, the proportion of population living in rural areas, population density and the primary gender parity index) to account for macro-level factors.

We used pre-pandemic data from 2019 (or the single most recent available year where data for 2019 were not available) for all health system and macro factors. We used data from a single year for these measures, as they are less susceptible to year-on-year variations.

**Data analyses**

We generated descriptive statistics using an exploratory data analysis approach to analyse the association between COVID-19 vaccination coverage and independent variables listed in Table 1. We calculated correlation coefficients where independent variables were continuous, as they allow an investigation into the strength and direction of an association between variables [42]. We generated Spearman correlations coefficients (as variables were non-normally distributed) with p-values ($p < 0.05$ considered significant) and 95% confidence intervals. We classified correlations as weak ($|r| < 0.4$),

moderate (|r| ≥ 0.4 and |r| < 0.7) or strong (|r| ≥ 0.7). We did not conduct linear regression analysis as there were substantial missing data from countries (especially non-WHO member states) which limited the number of countries that could be included. We also calculated COVID-19 vaccination coverage for all categorical variables.

We included data from SIDS that reported COVID-19 vaccination coverage data and were publicly available, excluding SIDS from analyses where data for specific independent variables were unavailable. In analyses with RI coverage, we used the mean of available coverage values (i.e., the denominator was the number of years for which data were available). We used 5-year mean annual immunisation coverage for RI rather than individual year estimates to account for year-to-year variation in coverage and mitigate against the impact of an anomalous year for a specific country in our analysis. We excluded SIDS from specific analyses only if they did not have any coverage values reported for a specific vaccine over the 5-year period. We limited the analyses by new vaccine introduction variables and immunisation system performance variables to WHO member states only (n = 39), as data for non-member states were not available through WUENIC.

Analyses were conducted in RStudio (R version 4.4.1, 2024) [43].

## Results

Of the 57 SIDS, two (Martinique and the US Virgin Islands) did not have publicly reported COVID-19 vaccination data and were excluded. Our final study dataset included 55 SIDS representing a mix of country income levels (see Table 2). The included SIDS were from the WHO regions of the Americas (n = 27), Western Pacific (n = 20), Africa (n = 6) and Southeast Asia (n = 2).

### Associations with routine immunisation performance

We found weak correlations (r < 0.4) between coverage of both the first dose and complete primary series of COVID-19 vaccination coverage and the 5-year mean annual coverage of RI included in the study, with few exceptions (Table 3; also see S1 Appendix). These exceptions where there were moderate correlations (i.e., r ≥ 0.4 but all were r < 0.5) were for coverage of the birth dose of hepatitis B vaccine (r = 0.425 [p = 0.006, 95%CI: 0.056–0.730] and r = 0.421 [p = 0.007, 95%CI: 0.072–0.721] for the first dose and full primary series of COVID-19 vaccination coverage, respectively, in June 2022, and r = 0.421 [p = 0.007, 95%CI: 0.072–0.721] and r = 0.438 [p = 0.005, 95%CI: 0.098–0.715] for the first dose and full primary series of COVID-19 vaccination coverage, respectively, in December 2022) and the first dose of measles vaccine (r = 0.420 [p = 0.002, 95%CI: 0.145–0.664] for full primary series of COVID-19 vaccination and MCV1 in December 2021). We observed moderate negative correlations (i.e., between –0.4 and –0.5) between COVID-19 vaccination coverage and dropout of coverage between routine vaccines, particularly between DTP1 and either MCV1 or MCV2, largely at the December 2021 timepoint (results in S2 Appendix). The negative correlations indicate lower COVID-19 vaccination coverage with higher RI dropout. Correlations at all other timepoints were weak.

We found SIDS that experienced declines in RI coverage in 2021 and 2022 had lower COVID-19 vaccination coverage across the entire study period compared with those who did not experience disruptions (see Fig 1). This was true whether SIDS experienced declines for a single vaccine (DTP1, DTP3 or MCV1) or all three combined vaccines.

### Association with new vaccine introductions

COVID-19 vaccination coverage was higher among SIDS that had introduced HPV, influenza and MCV2 vaccines. These differences were greater in the earlier stages of the COVID-19 vaccination rollout, and reduced over time. When examined by years of introduction, SIDS that had introduced HPV vaccine (see Fig 2) and MCV2 (see Fig 3) vaccine within 5 years of 2019 had similar levels of COVID-19 vaccination coverage to those who had not introduced those vaccines as of 2019. COVID-19 vaccination coverage was higher for SIDS that had introduced HPV vaccine and MCV2 for 5 or more years. The reverse was observed for influenza vaccine introduction, with higher COVID-19 vaccination coverage among SIDS

**Table 2. Characteristics of included small island developing states (SIDS) (N = 55)*.**

| Characteristic | n | % |
|---|---|---|
| WHO region | | |
| Africa | 6 | 10.9% |
| Americas | 27 | 49.1% |
| South-East Asia | 2 | 3.6% |
| Western Pacific | 20 | 36.4% |
| SIDS region | | |
| AIS | 8 | 14.5% |
| Caribbean | 27 | 49.1% |
| PICs | 19 | 34.5% |
| N/A | 1 | 1.8% |
| Income level | | |
| Low | 1 | 1.8% |
| Lower-middle | 11 | 20.0% |
| Upper-middle | 16 | 29.1% |
| High | 22 | 40.0% |
| Other | 5 | 9.1% |
| Status as "Least developed country" | | |
| Least developed | 7 | 12.7% |
| Developing | 48 | 87.3% |
| Eligibility for Gavi funding | | |
| Yes | 7 | 12.7% |
| No | 32 | 58.2% |
| N/A (non-WHO member states) | 16 | 29.1% |
| COVAX membership | | |
| Self-financing | 12 | 21.8% |
| AMC | 23 | 41.8% |
| No | 4 | 7.3% |
| N/A (non-WHO member states) | 16 | 29.1% |

AIS: Atlantic, Indian Ocean and South China Sea; AMC: Advance Market Commitment; COVAX: COVID-19 Vaccines Global Access; PICs: Pacific Island Countries and Territories; SIDS: Small Island Developing State(s); WHO: World Health Organization.

*The 55 SIDS included. American Samoa, Anguilla, Antigua and Barbuda, Aruba, Bahamas, Barbados, Belize, Bermuda, British Virgin Islands, Cabo Verde, Cayman Islands, Commonwealth of Northern Marianas, Comoros, Cook Islands, Cuba, Curacao, Dominica, Dominican Republic, Fiji, French Polynesia, Grenada, Guadeloupe, Guam, Guinea-Bissau, Guyana, Haiti, Jamaica, Kiribati, Maldives, Marshall Islands, Mauritius, Micronesia (Federated States of), Montserrat, Nauru, New Caledonia, Niue, Palau, Papua New Guinea, Puerto Rico, Saint Lucia, Saint Vincent and the Grenadines, Saint Kitts and Nevis, Samoa, São Tomé and Príncipe, Seychelles, Singapore, Sint Maarten, Solomon Islands, Suriname, Timor-Leste, Tonga, Trinidad and Tobago, Tuvalu, and Vanuatu.

that had introduced influenza vaccine within 5 years, noting there were only three countries in this group (see Fig 4). No trends were observed regarding introduction of PCV or rotavirus vaccine (see S3 Appendix).

## Association with health system performance

We found a strong correlation between first dose and full COVID-19 vaccination coverage and the density of physicians, particularly in 2021 (first dose – June 2021: 0.905; December 2021: 0.759; primary series – June 2021: 0.897; December 2021: 0.785, p<0.001 for all), which declined over the study period to a moderate correlation in 2022 (first dose – June 2022: 0.643, p=0.001; December 2022: 0.608, p=0.002; primary series – June 2022: 0.654, p=0.001; December 2022: 0.625, p=0.001

**Table 3. Spearman correlations between COVID-19 vaccination coverage and 5-year (2015–2019) mean annual coverage of routine immunisations.**

| Vaccine | N | June 2021 | December 2021 | June 2022 | December 2022 |
|---|---|---|---|---|---|
| | | rho | rho | rho | rho |
| **Coverage of first dose of COVID-19 vaccination** | | | | | |
| BCG | 42 | 0.335* | 0.279 | 0.257 | 0.231 |
| DTP1 | 51 | 0.334* | 0.35* | 0.334* | 0.314* |
| DTP3 | 52 | 0.264 | 0.348* | 0.258 | 0.222 |
| HepB birth dose | 40 | 0.367* | 0.374* | **0.425**** | **0.402*** |
| MCV1 | 52 | 0.338* | 0.399** | 0.341* | 0.304* |
| MCV2 | 47 | 0.245 | 0.377** | 0.338* | 0.317* |
| **Full coverage of primary series of COVID-19 vaccination** | | | | | |
| BCG | 42 | 0.309* | 0.294 | 0.242 | 0.230 |
| DTP1 | 51 | 0.311* | 0.392** | 0.360** | 0.345* |
| DTP3 | 52 | 0.255 | 0.378** | 0.311* | 0.250 |
| HepB birth dose | 40 | 0.279 | 0.369* | **0.421**** | **0.438**** |
| MCV1 | 52 | 0.267 | **0.420**** | 0.373** | 0.334* |
| MCV2 | 47 | 0.178 | 0.393** | 0.359* | 0.349* |

*p<0.05, **p<0.01.

Bolding shows show |r|>0.4.

(see Table 4 and S4 Appendix; S5 Appendixfor graphs). We observed a moderate correlation between first dose and full COVID-19 vaccination coverage and the density of nurses and midwives, which was more closely correlated in 2021 than in 2022 (see Table 4).

We did not observe any clear associations between COVID-19 vaccination coverage and other health systems variables included in this study, including having a functional NITAG, the proportion of births registered, UHC index, government expenditure on healthcare, and under-5 and infant mortality rates (see S6 and S7 Appendices). While some statistically significant moderate correlations were observed, visual inspection of scatter plots showed clustering of datapoints with correlations driven by a handful of datapoints (see S7 Appendix).

### Association with country-level characteristics (economic, development and demographic factors)

COVID-19 vaccination coverage was higher in SIDS with higher country income during the period from January 2021 to December 2022. Coverage was highest in high-income countries, followed by upper-middle, lower-middle and low-income countries (see Fig 5, and Table 5). COVID-19 vaccination coverage was lower among SIDS classified as a "least developed country" during the same period, and in those eligible to receive Gavi funding (Table 5; see S8 Appendix for country classifications). COVID-19 vaccination coverage was highest in the four SIDS who did not participate in the COVAX facility, followed by self-financing members and AMC members (Table 5).

No clear associations were observed between demographic factors including proportion of population in rural areas, population density and population size, or the primary gender parity index (see S9 Appendix).

### Discussion

Our analysis of immunisation in SIDS provides critical insights into the relationship between RI and vaccination during infectious disease epidemics. We found that sustaining pre-pandemic RI coverage rates during the pandemic was positively associated with having high COVID-19 vaccination coverage. Positive associations were also found between

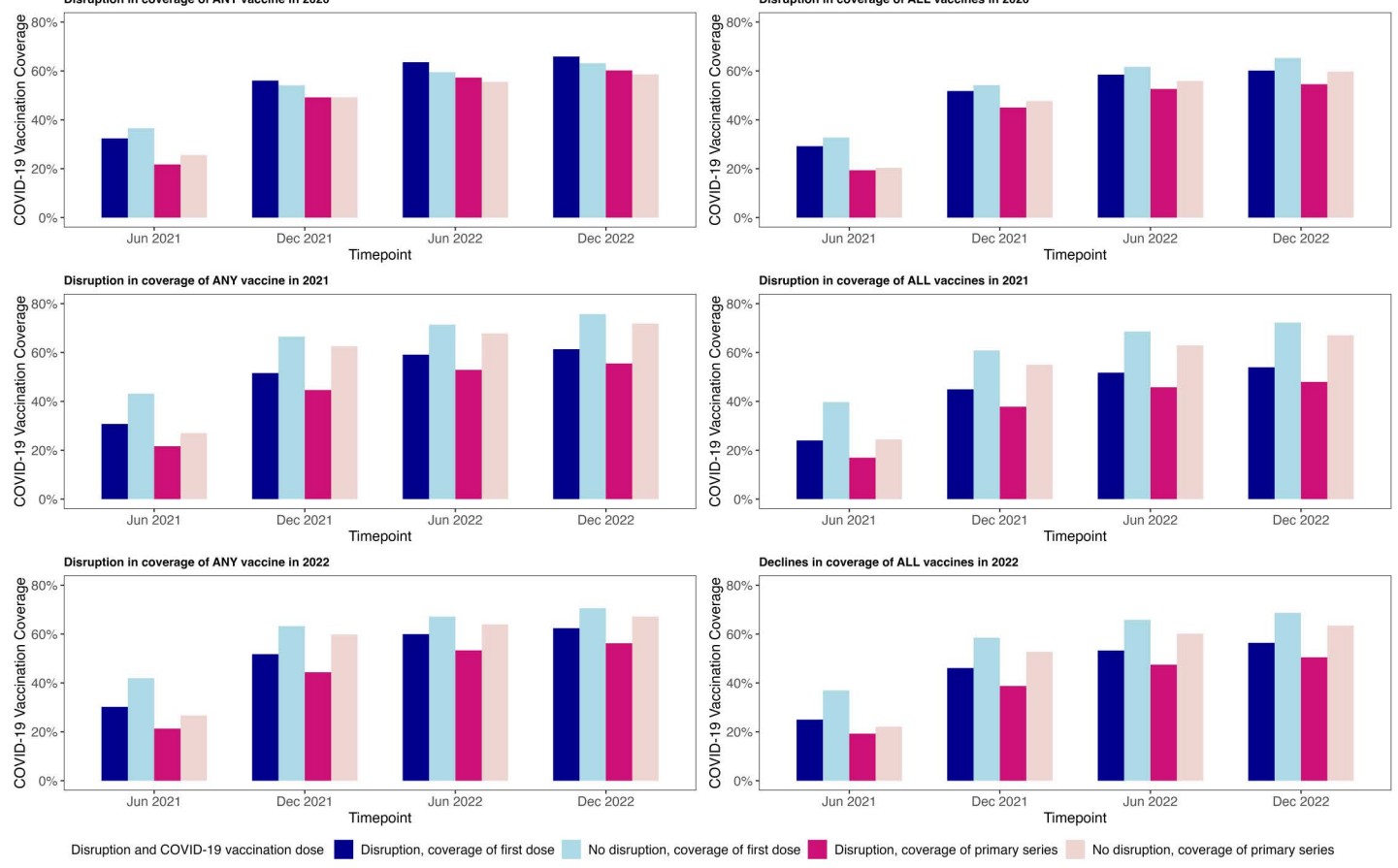

**Fig 1. COVID-19 vaccination coverage by whether countries experienced declines in routine immunisation coverage (DTP1, DTP3 and MCV1) during the COVID-19 pandemic.** The figures show COVID-19 vaccination coverage (first dose and primary series) at the four timepoints included in the study (i.e., June 2021, December 2021, June 2022 and December 2022), categorised by whether countries experienced disruptions to three routine immunisations (DTP1, DTP3 and MCV1) during the COVID-19 pandemic years (2020, 2021 and 2022).

Notes for Fig 1:
• Disruption in ANY vaccine: declines in coverage of at least one of the three vaccines (DTP1, DTP3 or MCV1) were observed in the given year.
• Disruption in ALL vaccines: declines in coverage of all three vaccines (DTP1, DTP3 or MCV1) were observed in the given year.
• DTP = diphtheria-tetanus-pertussis vaccine; MCV = measles-containing vaccine; number represents the dose number.

COVID-19 vaccination coverage and other immunisation and health system performance factors particularly health workforce density, having introduced newer vaccines, and economic and development factors. In contrast, there were mostly weak associations between COVID-19 vaccination coverage and RI coverage rates achieved prior to the COVID-19 pandemic in SIDS.

The weak associations between COVID-19 vaccination and level of RI coverage prior to the pandemic is not entirely surprising, since the speed, scale and target population of COVID-19 vaccination programs differed to those of RI programs which usually focusses on children under 5 years of age. It is consistent with results of earlier assessments which reported an absence of a relationship between national immunisation capacity and readiness to implement COVID-19 vaccination programs [44], and between the maturity of childhood immunisation programs and COVID-19 vaccination coverage [8]. The moderate associations that we observed between COVID-19 vaccination and RI vaccination coverage were largely with coverage of the birth dose of hepatitis B vaccine. This is typically given in a health facility post-delivery, which

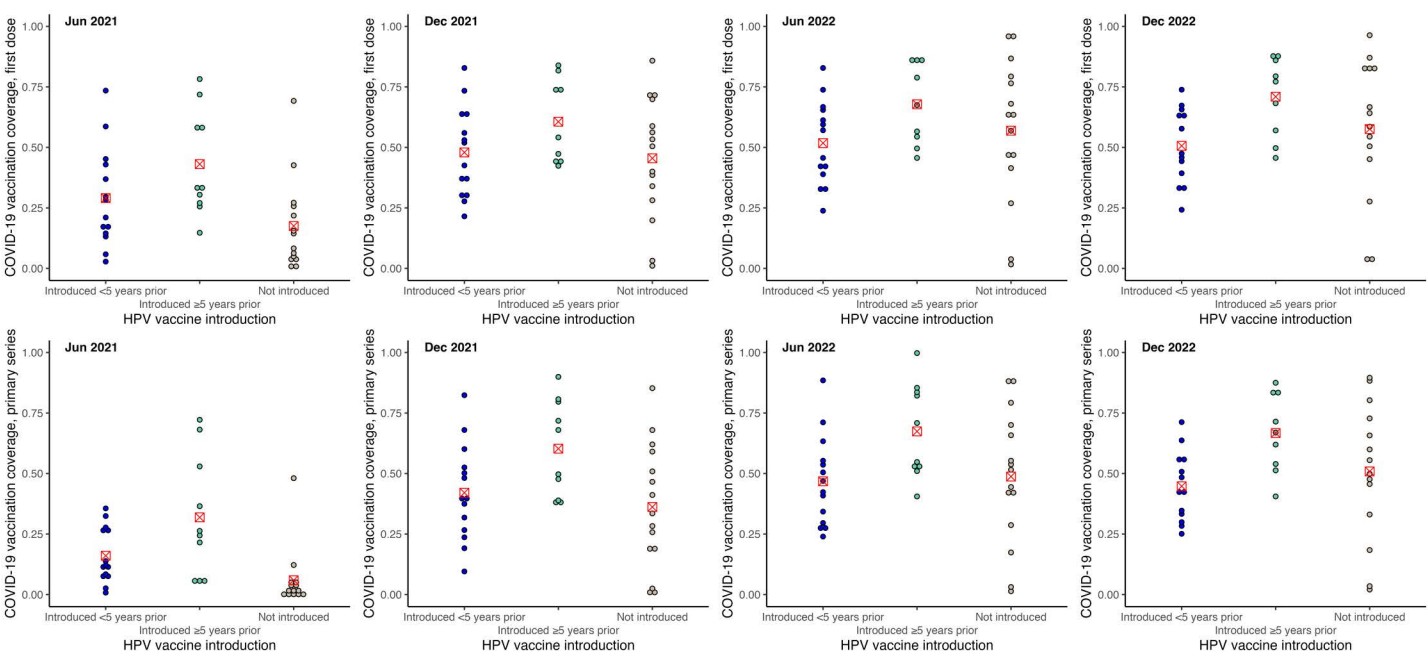

**Fig 2. COVID-19 vaccination coverage by status of HPV vaccine introduction.** Figures show COVID-19 vaccination by new vaccine introduction status: 1) introduced <5 years, 2) introduced ≥5 years, 3) not introduced. Dots represent COVID-19 vaccination coverage in each country, rounded to the nearest coverage point (e.g., 91.7% is rounded to 92%). The red marker represents mean COVID-19 vaccination coverage for the vaccine introduction category.

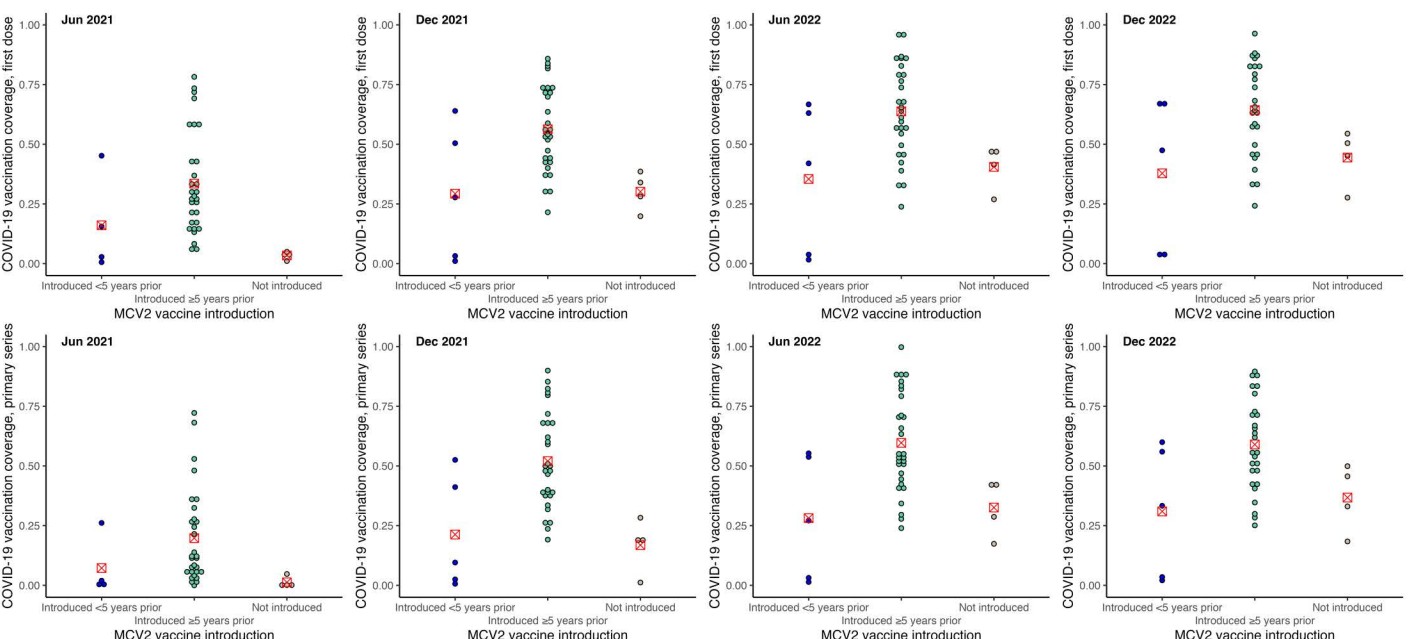

**Fig 3. COVID-19 vaccination coverage by status of measles-containing vaccine (second dose, MCV2) introduction.** Figures show COVID-19 vaccination by new vaccine introduction status: 1) introduced <5 years, 2) introduced ≥5 years, 3) not introduced. Dots represent COVID-19 vaccination coverage in each country, rounded to the nearest coverage point (e.g., 91.7% is rounded to 92%). The red marker represents mean COVID-19 vaccination coverage for the vaccine introduction category.

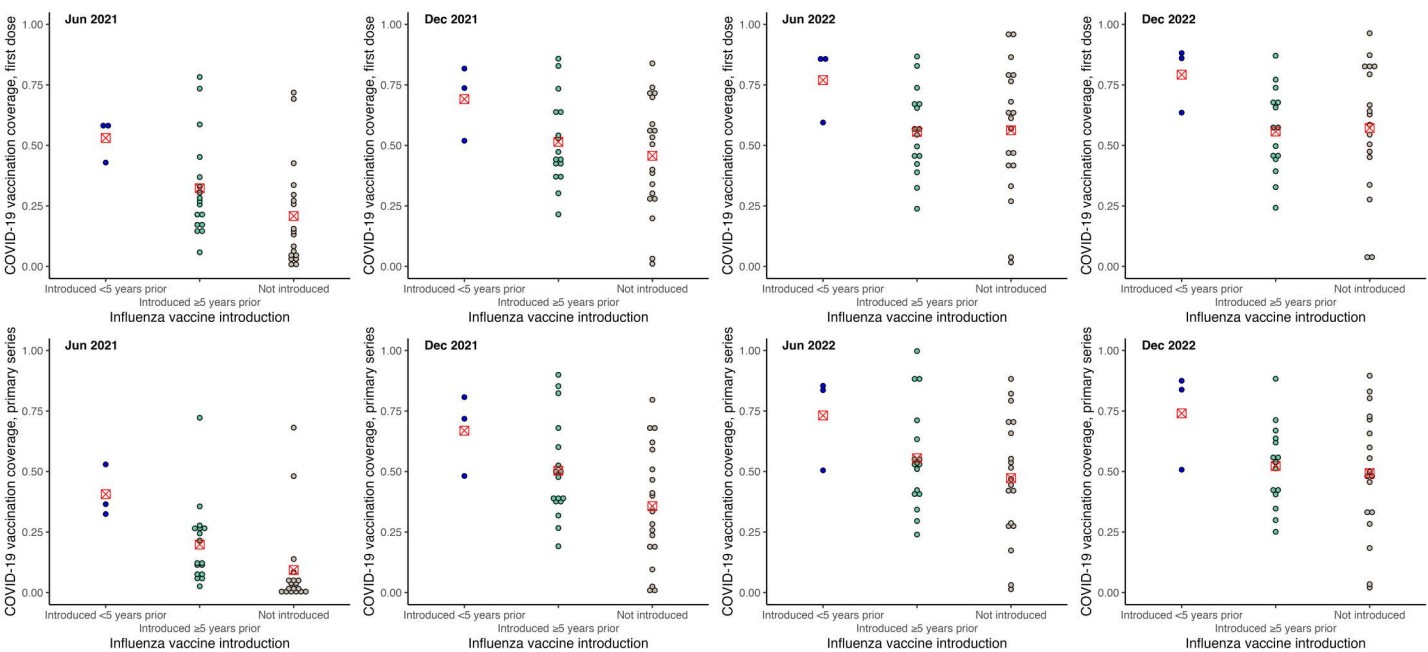

**Fig 4. COVID-19 vaccination coverage by status of influenza vaccine introduction.** Figures show COVID-19 vaccination by new vaccine introduction status: 1) introduced <5 years, 2) introduced ≥5 years, 3) not introduced. Dots represent COVID-19 vaccination coverage in each country, rounded to the nearest coverage point (e.g., 91.7% is rounded to 92%). The red marker represents mean COVID-19 vaccination coverage for the vaccine introduction category.

**Table 4. Spearman correlations between COVID-19 vaccination coverage and density of health resources (workforce and hospital beds).**

| Vaccine | June 2021 | December 2021 | June 2022 | December 2022 |
|---|---|---|---|---|
| | rho | rho | rho | rho |
| **Coverage of first dose of COVID-19 vaccination** | | | | |
| Physicians per 1,000# | **0.905*** | **0.759*** | **0.643** | **0.608** |
| Nurses and midwives per 1,000 | **0.598** | **0.517*** | 0.389 | 0.360 |
| Hospital beds per 10,000 | **0.581*** | 0.312 | 0.290 | 0.297 |
| **Full coverage of primary series of COVID-19 vaccination** | | | | |
| Physicians per 1,000* | **0.897*** | **0.785*** | **0.654** | **0.625** |
| Nurses and midwives per 1,000 | **0.630** | **0.605** | 0.429* | 0.430* |
| Hospital beds per 10,000 | **0.641*** | 0.370* | 0.314 | 0.315 |

*p < 0.05, **p < 0.01, ***p < 0.001.

Bolding shows show |r| > 0.4.

Shading and folding shows strong correlations, i.e., |r| > 0.7.

#One country was an outlier and excluded from this analysis. Correlation values were strong in a sensitivity analysis that included the outlier country.

differs from vaccines given at other timepoints in infancy which are delivered through a combination of fixed, outreach and mobile vaccination services, implying access to health facilities may have been an important factor for attaining COVID-19 vaccination coverage in SIDS. We also observed moderate negative correlations between COVID-19 vaccination coverage and dropout of annual coverage for RI in December 2021, when COVID-19 vaccination supply constraints were

**A. Monthly COVID-19 vaccination coverage, by country income level**

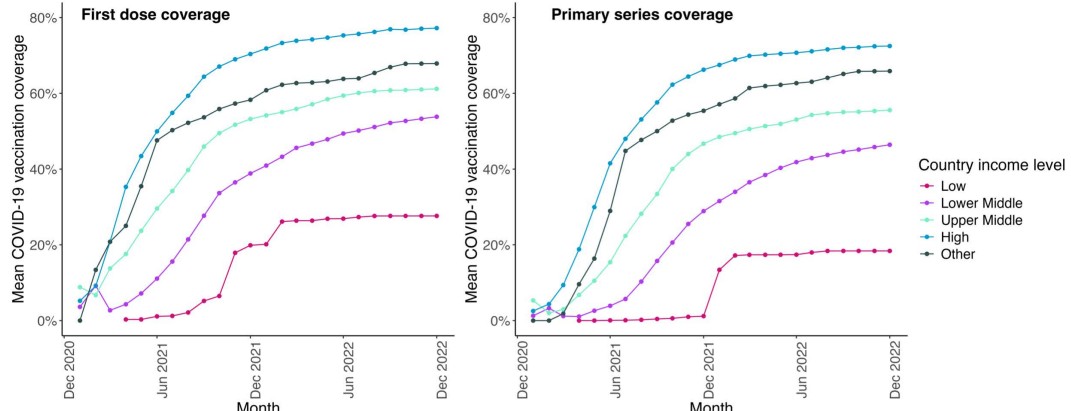

**B. Monthly COVID-19 vaccination coverage, by classification as a least developed country**

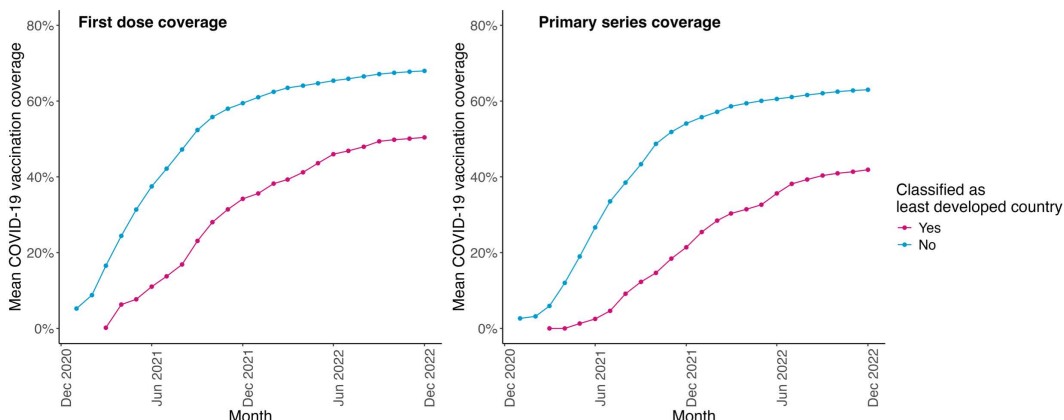

**C. Monthly COVID-19 vaccination coverage, by eligibility to receive Gave funding**

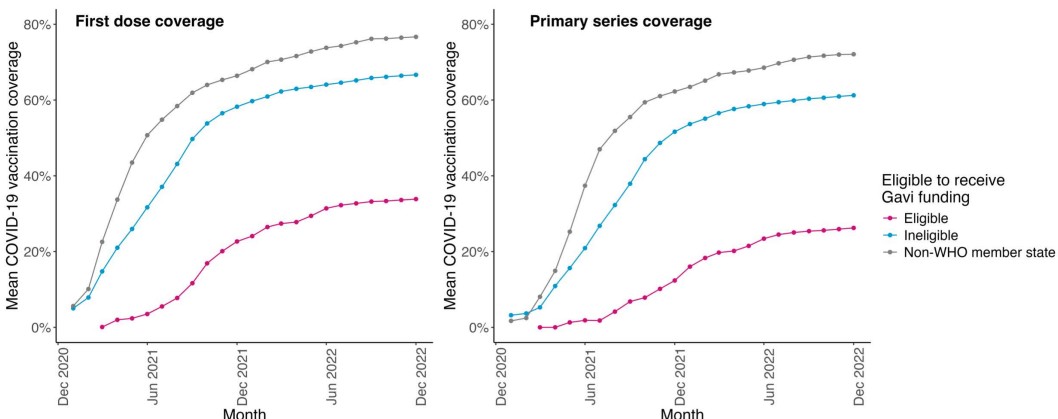

**Fig 5. Monthly COVID-19 vaccination coverage by economic factors.** Figures show monthly COVID-19 vaccination coverage (first dose and primary series) by various economic factors, namely country income level, status as a least developed country, and eligibility for Gavi funding.

**Table 5. COVID-19 vaccination coverage at four timepoints in 2021 and 2022, by economic factors.**

| Characteristic | N | Coverage of first dose of COVID-19 vaccination | | | | Coverage of primary series of COVID-19 vaccination | | | |
|---|---|---|---|---|---|---|---|---|---|
| | | Jun 2021 | Dec 2021 | Jun 2022 | Dec 2022 | Jun 2021 | Dec 2021 | Jun 2022 | Dec 2022 |
| Income level | | | | | | | | | |
| Low | 1 | 1.1% | 19.9% | 26.9% | 27.6% | 0.1% | 1.2% | 17.4% | 18.4% |
| Lower-middle | 11 | 10.8% | 38.5% | 49.4% | 54.2% | 3.9% | 27.9% | 42.0% | 46.8% |
| Upper-middle | 16 | 29.4% | 53.4% | 60.0% | 61.6% | 16.2% | 47.5% | 53.5% | 56.1% |
| High | 22 | 47.6% | 68.3% | 73.2% | 75.7% | 38.7% | 63.6% | 68.5% | 70.6% |
| Other | 5 | 47.6% | 58.3% | 63.8% | 67.9% | 29.0% | 55.4% | 62.7% | 65.9% |
| LDC status | | | | | | | | | |
| Yes | 7 | 11.0% | 34.2% | 46.0% | 50.4% | 2.5% | 21.4% | 35.6% | 41.9% |
| No | 48 | 37.5% | 59.4% | 65.4% | 68.0% | 26.7% | 54.1% | 60.6% | 62.9% |
| Eligibility for Gavi funding | | | | | | | | | |
| Yes | 7 | 3.5% | 22.7% | 31.4% | 33.8% | 1.9% | 12.4% | 23.4% | 26.2% |
| No | 31 | 31.7% | 58.2% | 64.1% | 66.7% | 20.9% | 51.6% | 58.9% | 61.2% |
| N/A* | 17 | 50.7% | 66.4% | 73.8% | 77.2% | 37.4% | 62.3% | 68.5% | 72.4% |
| COVAX status | | | | | | | | | |
| AMC | 23 | 18.7% | 42.7% | 52.1% | 55.6% | 8.4% | 33.6% | 44.4% | 48.4% |
| Self-financing | 12 | 37.2% | 57.8% | 63.4% | 65.9% | 24.8% | 53.5% | 58.1% | 61.0% |
| No | 4 | 56.3% | 81.6% | 85.4% | 90.4% | 36.1% | 79.8% | 85.6% | 89.1% |
| N/A | 16 | 48.9% | 68.2% | 72.5% | 74.0% | 41.8% | 63.2% | 68.5% | 69.5% |

LDC = Least Developed Country.

*Non-member WHO states.

easing. This may indicate the role of disparities in wealth, education and other socioeconomic factors, which are predictors of vaccination dropout, in achieving high and equitable coverage during an epidemic [45].

We found that SIDS that did not experience declines in RI coverage during the pandemic also achieved high COVID-19 vaccination coverage. This supports the principle that a "strong" health system is a resilient one. The absence of a decline in RI coverage observed in these countries does not imply that there were no disruptions to immunisation services during the pandemic. Rather, it is more likely that they were able to reorganise and rapidly recover from any disruptions. This ability to rapidly recover from shocks to the system while responding to the crisis is a defining characteristic of resilient health systems [46].

Resilience is dependent on having a high functioning and adaptive health system, underpinned by equitable access to essential health services and a skilled and adequately sized workforce [46]. While higher COVID-19 vaccination coverage was correlated with the workforce density and having introduced and sustained newer vaccines, we did not find an association between universal health coverage and COVID-19 vaccination coverage. This is contrary to a recent study that demonstrated countries with higher universal health coverage index scores had higher rates of COVID-19 vaccination [47]. Our findings might reflect intensified international development partner efforts in response to COVID-19 especially in lower income countries, or the public response to COVID-19 control measures, such as requirements for vaccination certificates for travel [48].

The association between COVID-19 vaccination coverage and the density of physicians was the strongest in our analysis, with a weaker association with the density of nurses and midwives observed. This adds to the literature on the importance of having sufficient skilled health professionals to meet the surge requirements of a health emergency, contributing to the case for ongoing investment in building the health workforce. Having an adequate number of skilled health workers enables task shifting and redistributing workers in an emergency response while mitigating the impact on routine services, leading to resilience [49].

Limited workforce quantity and capability were key barriers in COVID-19 vaccination program implementation globally [50,51]. Many countries had to diversify their workforce by recruiting students and retired health professionals [52]. Modelling studies found an additional 744,000 health workers were needed globally, largely in low-income countries, to achieve 70% coverage for COVID-19 vaccination in all countries by mid-2022 at a cost of US$2.5 billion [51]. While workforce capacity is critical for health system resilience in all settings, it is possibly more important in SIDS where the small numbers of health professionals greatly limits surge capacity, [11,13,17] and where workers take on multiple roles even under normal circumstances [53]. The need to invest in building the health workforce in SIDS is recognised by target 3c of the Sustainable Development Goals, which is to "substantially increase health financing and the recruitment, development, training and retention of the health workforce in developing countries, especially in least developed countries and small island developing states" [54]. In these settings, a small number of additional health workers could have impacted countries' ability to deliver immunisation services.

Similar to other studies, [55–58] we found a clear association between COVID-19 vaccination coverage and country income level. In our study, least developed countries, with weaker health systems, [59] had lower COVID-19 vaccination coverage throughout the study period. It also reflects procurement and access to COVID-19 vaccines with low-and-middle-income countries unable to out-compete wealthy countries in negotiations with manufacturers [56]. For SIDS, vaccine logistics are more challenging and expensive due to geographically remote and dispersed islands. Small population sizes in SIDS also diminishes their purchasing power [12,60]. We found inequities in coverage persisted throughout 2022 when COVID-19 vaccine supply was no longer an issue, indicating health systems factors other than purchasing power and access to vaccine supplies contribute and are critical for vaccine delivery during epidemics. This includes workforce density, which is associated with country income [51].

The relationship with introduction of new vaccines (e.g., influenza, HPV and MCV2) aligns with previously published data that showed an association between COVID-19 vaccination coverage and having an adult seasonal influenza program [8]. New vaccine introduction requires all components of the immunisation system to work together, including policy decision financing and vaccine procurement, cold chain capacity, workforce training and community engagement [61,62]. Furthermore, the associations we found were with vaccines introduced to age groups beyond the first year of life. This indicates that system strengthening occurs through the introduction of new systems and processes needed to deliver vaccines to a target population other than infants, and especially when the target population is in adolescence or adulthood. The benefits of new vaccine introductions such as increased reach and integration of health services following new service delivery mechanisms, increased workforce skill and strengthening adverse event surveillance have been described qualitatively, [38,62,63] but are challenging to quantify. Our study provides evidence that the increased flexibility, adaptability and capacity of the health system following new vaccine introductions is linked to achieving higher coverage of vaccination in an emergency scenario. Our finding regarding the relationship with years since introduction suggests that it takes time for countries to adjust and embed new vaccine delivery processes into health systems.

Our analysis was limited by the lack of publicly available coverage data for all routinely-administered vaccines in SIDS, particularly for non-WHO member states. Inconsistencies in annual reporting and variation in the quality of immunisation coverage data across SIDS are potential sources of bias. These biases are mitigated in part through the longstanding standardised process of reporting immunisation coverage estimates through the JRF, a process that has been evaluated and validated [64]. A recent assessment showed that data quality has improved between 2000 and 2019 [65]. We also did not examine inequities in vaccination coverage, for COVID-19 or routine vaccinations, within countries, due to the lack of publicly available data, poor quality of subnational level data and challenges in interpreting variations in coverage proportions in countries with small and highly mobile populations [66]. Due to the small number of SIDS in some categories, we were unable to conduct regression analyses and control for potential economic and demographic confounders such as country income level and population size. While this study is unable to draw conclusions about causal associations between routine immunisation systems and COVID-19 vaccination coverage, it does reveal clear links between emergency vaccination and health and immunisation systems in SIDS.

While our study focussed on disruptions to RI, future research to identify and measure health system factors predictive of resilient immunisation systems would be useful. Additional factors that could be examined include service delivery and accessibility, vaccine acceptance and demand, and cold chain capacity, which were beyond the scope of this study. Cultural and political factors also have a role, with multiple studies reporting that trust in government was strongly associated with higher, faster and earlier uptake of COVID-19 vaccination and other outcomes during the pandemic including COVID-19 mortality [8,67,68]. Other factors, such as policies related to mandatory vaccination requirements for work or travel, geopolitical factors including vaccine diplomacy, and the role and influence of external donors and global health partner organisations could have also influenced COVID-19 vaccine uptake. Qualitative methods could be used to explore other health systems characteristics such as flexibility and adaptability, which are difficult to quantify.

## Conclusions

In conclusion, we found that countries that achieved higher COVID-19 vaccination coverage also sustained pre-pandemic RI coverage during the pandemic, had introduced newer vaccines into their national immunisation schedules, and had higher workforce density. Our study provides support that high-performing health systems are resilient in achieving public health outcomes like high vaccination coverage during an epidemic while sustaining routine vaccination. They are underpinned by a health workforce, sufficient in both quantity and capability, and a flexible system able to adapt existing service delivery models quickly to meet emergency needs. Our study provides insights on where system strengthening efforts can focus to prevent shocks to health services not just for immunisation but primary health care in general. We recommend that countries continue to invest in their health workforce including those working in immunisation. Increasing efforts to introduce new vaccines into national immunisation schedules, including implementing new service delivery mechanisms to reach populations beyond the first year of life, can increase adaptability and thus place countries in a better position to deploy epidemic and pandemic vaccination programs.

## Supporting information

**S1 Appendix. Spearman correlations between COVID-19 vaccination coverage and 5-year (2015–2019) mean annual coverage of routine immunisations.**
(PDF)

**S2 Appendix. Spearman correlations between COVID-19 vaccination coverage and dropout of annual coverage (5-year mean, 2015–2019) of routine immunisations.**
(PDF)

**S3 Appendix. COVID-19 vaccination coverage by new vaccine introductions.**
(PDF)

**S4 Appendix. Spearman correlations between COVID-19 vaccination coverage and density of health resources (workforce and hospital beds).**
(PDF)

**S5 Appendix. Scatterplots of COVID-19 vaccination coverage and workforce density.**
(PDF)

**S6 Appendix. COVID-19 vaccination coverage at four timepoints in 2021 and 2022, by health system variables included in the study.**
(PDF)

**S7 Appendix. Scatterplots of COVID-19 vaccination coverage and health system variables included in the study.**
(PDF)

**S8 Appendix. List of countries categorised by economic factor variables.**
(PDF)

**S9 Appendix. Scatterplots of COVID-19 vaccination coverage and demographic factors included in the study.**
(PDF)

## Acknowledgments

We would like to acknowledge the statistical training services provided by Biological Data Science Institute at the Australian National University and Sydney Informatics Hub at the University of Sydney. CP is supported by an Australian Government Research Training Program (RTP) Scholarship.

## Author contributions

**Conceptualization:** Cyra Patel, Ginny Sargent, Meru Sheel.

**Data curation:** Cyra Patel, Gizem Bilgin.

**Formal analysis:** Cyra Patel, Gizem Bilgin.

**Investigation:** Cyra Patel.

**Methodology:** Cyra Patel, Andrew Hayen, Aditi Dey, Meru Sheel.

**Project administration:** Cyra Patel.

**Supervision:** Martyn Kirk, Ginny Sargent, Meru Sheel.

**Visualization:** Cyra Patel.

**Writing – original draft:** Cyra Patel.

**Writing – review & editing:** Cyra Patel, Gizem Bilgin, Andrew Hayen, Martyn Kirk, Akeem Ali, Aditi Dey, Ginny Sargent, Meru Sheel.

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
