## [Decision Letter · Decision Letter 0]

Dear Dr. Patel,

Thank you for submitting your manuscript to PLOS ONE. After careful consideration, we feel that it has merit but does not fully meet PLOS ONE’s publication criteria as it currently stands. Therefore, we invite you to submit a revised version of the manuscript that addresses the points raised during the review process.

We look forward to receiving your revised manuscript.

Kind regards,

Harapan Harapan, MD, PhD

Academic Editor

PLOS ONE

**Journal Requirements:**

Reviewers' comments:

Reviewer's Responses to Questions

**Comments to the Author**

1. Is the manuscript technically sound, and do the data support the conclusions?

Reviewer #1: No

Reviewer #2: Partly

2. Has the statistical analysis been performed appropriately and rigorously?

Reviewer #1: No

Reviewer #2: No

3. Have the authors made all data underlying the findings in their manuscript fully available?

Reviewer #1: No

Reviewer #2: Yes

4. Is the manuscript presented in an intelligible fashion and written in standard English?

Reviewer #1: Yes

Reviewer #2: No

**Reviewer #1:**  Tables 1 and 2 provide independent variables and characteristics of the study sample.

Table 3: Spearman correlations between COVID-19 vaccination coverage and 5-year (2015–2019) mean annual coverage of routine Immunisations.

What is the point of studying the correlation between COVID-19 vaccination coverage and 5-year (2015–2019) mean annual coverage of routine Immunisations?

Table 4: Spearman correlations between COVID-19 vaccination coverage and density of health resources (workforce and hospital beds)

I do not see any critical points from the data provided in Table 4

Table 5: COVID-19 vaccination coverage at four time points in 2021 and 2022, by economic factors.

I do not think it will be beneficial to do these comparisons. Instead, it would be a good idea to specify the countries in names instead of this classification.

In total, I do not believe this manuscript will add valuable information to the field.

**Reviewer #2: ** The manuscript with the title "The association between routine immunisation and COVID-19 vaccination in small island developing states" aimed to understanding the link between routine immunisation (RI) performance and vaccination during an epidemic. The study used multiple data sources (i.e., WHO/UNICEF Joint Reporting Form, World Bank Indicators) to enhance the robustness of the findings. In addition, the study collected information on health system factors, economic indicators, and demographic characteristics, allows for a multifaceted analysis. Below are the comments need to be addressed before consideration for publication.

1. The study depends primarily on Spearman correlations, which cannot prove causation. The analysis limits the capacity to draw firm conclusions regarding causative pathways. The study lacks extensive statistical modelling (e.g., regression analysis).

2. Inconsistent data from non-WHO member states might limits the findings' applicability to all SIDS.

3. The use of publicly available datasets may result in biases or inaccuracies due to different reporting standards and operation standards.

4. The study acknowledges the impact of health system resilience but does not delve into other important elements such community trust, vaccine hesitancy, and political pressures.

5. Another limitation of the current is that the study does not account for potential lag effects between routine immunization performance and COVID-19 vaccination uptake.

6. The study mentions supply issues. There are other important factors which are not considered in the current study including how geopolitical factors (e.g., vaccine diplomacy, reliance on external donors) impacted vaccine access in SIDS.

7. The study depends on countries with good reporting system for Routine Immunisation.

8. The focus of this study is on aggregate country-level statistics does not consider within-country differences, which may be considerable in SIDS.

9. The study shall provide more specific recommendation based on the findings, particularly in resource-limited areas.

**Do you want your identity to be public for this peer review?** For information about this choice, including consent withdrawal, please see our Privacy Policy

Reviewer #1: **Yes: ** Omeed Darweesh

Reviewer #2: **Yes: ** YIN CHENG LIM

---

## [Author Response · Author response to Decision Letter 1]

6 Mar 2025

We thank the reviewers for the comments on our paper and for the opportunity to address them. We have clarified certain aspects of our paper and included additional details in the manuscript in response. Below, we provide responses to each of the reviewers’ comments and line number references where we have made changes, or point to where existing text in the manuscript addresses comments.

Responses to reviewer 1

Comment 1: Tables 1 and 2 provide independent variables and characteristics of the study sample.

Table 3: Spearman correlations between COVID-19 vaccination coverage and 5-year (2015–2019) mean annual coverage of routine Immunisations.

What is the point of studying the correlation between COVID-19 vaccination coverage and 5-year (2015–2019) mean annual coverage of routine Immunisations?

Response: We analysed the correlation between countries’ 5-year mean routine vaccination coverage and COVID-19 vaccine coverage to answer our research question of whether there is a relationship between a robust childhood immunisation system and having higher coverage of vaccination delivered in response to an infectious disease outbreak. We have clarified why we have used correlation statistics in the text added to the methods section.

Changes to manuscript: Methods: data analyses section, page 15, line 259–260, text added:

We calculated correlation coefficients where independent variables were continuous, as they allow an investigation into the strength and direction of an association between variables. (underlined text is additional)

Comment 2: Table 4: Spearman correlations between COVID-19 vaccination coverage and density of health resources (workforce and hospital beds).

I do not see any critical points from the data provided in Table 4.

Response: We have expanded our discussion to more explicitly examine the results presented in Table 4. In our analysis, the correlation between COVID-19 vaccination coverage and physician density was the strongest (r=0.905 for June 2021, r=0.759 for December 2021 for coverage of the first dose of COVID-19 vaccination; r=0.897 for June 2021, r=0.785 for December 2021 for coverage of the complete primary series of COVID-19 vaccination). We also found a moderate correlation between the density of nurses and midwives and COVID-19 vaccination coverage. These findings are mentioned on page 22, lines 375–383. We discuss the significance of these findings in the discussion section (paragraph starting on page 28, line 472), i.e. that this demonstrates the relationship between the health workforce and delivery of vaccination services during a public health emergency. This is particularly important in small island country settings where workforce constraints are even more pronounced. Our findings add to the case for investment in the health workforce, both in quantity and quality to ensure effective and strong vaccination programs during emergencies.

Changes to manuscript: Discussion section, page 28, lines 474–478, text added:

This adds to the literature on the importance of having sufficient skilled health professionals to meet the surge requirements of a health emergency, contributing to the case for ongoing investment in building the health workforce. Having an adequate number of skilled health workers enables task shifting and redistributing workers in an emergency response while mitigating the impact on routine services, leading to resilience. (underlined text is additional)

Discussion section, page 28–29, lines 487–493 text added:

The need to invest in building the health workforce in SIDS is recognised by target 3c of the Sustainable Development Goals, which is to “substantially increase health financing and the recruitment, development, training and retention of the health workforce in developing countries, especially in least developed countries and small island developing states”.

Comment 3: Table 5: COVID-19 vaccination coverage at four time points in 2021 and 2022, by economic factors.

I do not think it will be beneficial to do these comparisons. Instead, it would be a good idea to specify the countries in names instead of this classification.

Response: We examined COVID-19 vaccination coverage at four timepoints during the acute phase of the pandemic to examine if associations with independent variables held across all timepoints or if they changed over the 2-year period. Supply constraints earlier in the global rollout (especially in 2021) affected several countries’ ability to scale up their programs. Other factors, such as new COVID-19 variants, changing policies on infection control measures and their impact on the health system would have affected capacity to deliver vaccination programs.

Country-level wealth was extremely influential in access to COVID-19 vaccines during the pandemic, and thus was expected to impact COVID-19 vaccination coverage. This has been demonstrated in other studies, including:

• Nabaggala et al. Int J Equity Health. 2022;21: 147. doi:10.1186/s12939-022-01750-0

• Glassman et al. COVID-19 Vaccine Development and Rollout in Historical Perspective. Washington DC: Center for Global Development Working Paper 607; 2022. Available from: https://www.cgdev.org/sites/default/files/covid-19-vaccine-development-and-rollout-in-historical-perspective-paper.pdf

An individual-level analysis would prevent a clear examination of the relationship between COVID-19 vaccination coverage and economic factors in small island developing states. Instead, we have added an appendix that lists which countries fall under each category for the four variables we have included in our analysis.

Changes to manuscript: We have included an appendix listing which countries were included in each category for economic factors. Please see appendix S8.

Comment 4: In total, I do not believe this manuscript will add valuable information to the field.

Response: We thank the reviewer for their feedback. We have expanded the study rationale to more clearly describe our study’s contribution to the field.

Changes to manuscript:

Text added to introduction, page 4, lines 96–101, in relation to what the study adds to the literature:

However the study did not examine factors such as the ability to maintain routine immunisation coverage while concurrently implementing COVID-19 vaccination programs (a measure of system resilience), universal health coverage index and infant mortality. These indicators measure different aspects of health and immunisation system performance and including them such analyses provides additional insights into how strengthening routine systems can improve outcomes during an emergency.

Added text to the discussion section:

Page 27–28, lines 456–464, regarding findings related to disruptions to RI and resilience:

The absence of a decline in RI coverage observed in these countries does not imply that there were no disruptions to immunisation services during the pandemic, but more likely that they were able to reorganise and rapidly recover from any disruptions. This ability to respond to and rapidly recover to shocks to the system while responding to the crisis is a defining characteristic of resilient health systems.[41] Resilience is dependent on having a high functioning and adaptive health system, underpinned by equitable access to essential health services and a skilled and adequately sized workforce.[41] Our findings that higher COVID-19 vaccination coverage was correlated with the workforce density and having introduced and sustained newer vaccines reinforce this.

Page 28–29, lines 474–478 and 487–493, regarding findings related to workforce:

This adds to the literature on the importance of having sufficient skilled health professionals to meet the surge requirements of a health emergency, contributing to the case for ongoing investment in building the health workforce. Having an adequate number of skilled health workers enables task shifting and redistributing workers in an emergency response while mitigating the impact on routine services, leading to resilience... The need to invest in building the health workforce in SIDS is recognised by target 3c of the Sustainable Development Goals, which is to “substantially increase health financing and the recruitment, development, training and retention of the health workforce in developing countries, especially in least developed countries and small island developing states”.

Page 30, lines 518–524, regarding findings related to new vaccine introductions:

The benefits of new vaccine introductions such as increased reach of health services following new service delivery mechanisms, increased workforce skill and strengthening adverse event surveillance have been described qualitatively, but are challenging to quantify. Our study provides evidence that the increased flexibility, adaptability and capacity of the health system following new vaccine introductions is linked to achieving higher coverage of vaccination in an emergency scenario.

Responses to Reviewer 2

Overall comment: The manuscript with the title "The association between routine immunisation and COVID-19 vaccination in small island developing states" aimed to understanding the link between routine immunisation (RI) performance and vaccination during an epidemic. The study used multiple data sources (i.e., WHO/UNICEF Joint Reporting Form, World Bank Indicators) to enhance the robustness of the findings. In addition, the study collected information on health system factors, economic indicators, and demographic characteristics, allows for a multifaceted analysis. Below are the comments need to be addressed before consideration for publication.

Response: We thank the reviewer for their thoughtful comments, incorporating their feedback has strengthened our paper’s discussion (see below).

Comment 1: The study depends primarily on Spearman correlations, which cannot prove causation. The analysis limits the capacity to draw firm conclusions regarding causative pathways. The study lacks extensive statistical modelling (e.g., regression analysis).

Response: We agree that Spearman correlations do not prove causation. However, our intention was to examine the relationship between routine immunisation systems and emergency vaccination. As correlation coefficients describe the strength and direction of an association between variables, we believe these are appropriate statistical measures to achieve our research objective.

We were unable to conduct regression analyses due to limitations in the available data, namely missing data. For example, 13 of 55 SIDS (23.6%) had not reported any data on BCG vaccination coverage for the 5-year period from 2015 to 2019. Fifteen (27.3%) did not report any data on hepatitis B birth dose vaccination coverage for the same period. Data on new vaccine introductions was only available for SIDS who were WHO member states (39/55, 70.9%). Data on various health and immunisation system indicators was also absent, e.g. only 26/55 SIDS (47.3%) had indicated whether they had a system for monitoring adverse events following immunisations.

Changes to manuscript: Discussion section edited to acknowledge limitations regarding statistical approach, page 31, lines 541–544, text added:

While this study is unable to draw conclusions about causal associations between routine immunisation systems and COVID-19 vaccination coverage, it does reveal clear links between emergency vaccination and health and immunisation systems in SIDS.

Comment 2: Inconsistent data from non-WHO member states might limits the findings' applicability to all SIDS.

Response: We agree that inconsistent reporting of data from non-WHO member states may limit our findings’ applicability and have included this limitation in the discussion (page 30, from line 528). We expect country-by-country variation, however we consider our overall conclusions to be applicable to SIDS.

Changes to the manuscript: Discussion section, pages 30, lines 529–534, text added:

Inconsistencies in annual reporting and variation in the quality of immunisation coverage data across countries SIDS are potential sources of bias. These biases are mitigated in part through the longstanding standardised process of reporting immunisation coverage estimates through the JRF, a process that has been evaluated and validated.[59] A recent assessment showed that data quality has improved between 2000 and 2019.[60]

Comment 3: The use of publicly available datasets may result in biases or inaccuracies due to different reporting standards and operation standards.

Response: We acknowledge that there are limitations in the quality of the data (as with most datasets). However, immunisation data collected through the WHO/UNICEF Joint Reporting Form process are highly standardised through a process in place since 1998. All WHO member states report immunisation data annually, which are validated and adjusted by the WHO and UNICEF based on various data sources including surveys. The resulting WUENIC estimate of coverage is the vaccine coverage statistic which we have used in our study. These data have been validated in evaluations. See: Danovaro-Holliday et al. Gates Open Res. 2021;5:77. doi: 10.12688/gatesopenres.13258.1.

Changes to manuscript: Edited as above in response to comment #2.

Comment 4: The study acknowledges the impact of health system resilience but does not delve into other important elements such community trust, vaccine hesitancy, and political pressures.

Response: We acknowledge that our study did not explore all factors which may influence resilience. Our study focuses on measures related to overall immunisation system performance, especially vaccination coverage and introduction of new vaccines which are well-established measures of immunisation system performance (see Patel et al. Glob Health Sci Pract. 2023;11(3):e220055). We have clarified this in our methods section. Data on the factors mentioned by the reviewer is also limited and is not systematically collected, particularly for SIDS. This is noted in the discussion of study limitations (page 31, lines 547–555).

Changes to manuscript: Methods section (independent variables: routine immunisation), page 13, lines 212–214:

In this study, we focused on well-established measures of the overall performance of immunisation systems at a national level, namely indicators for vaccination coverage and introducing new vaccines in national immunisation programs.

No additional changes made to other sections – factors mentioned by reviewer are already noted in discussion of study limitations (page 31, lines 547–555).

Comment 5: Another limitation of the current is that the study does not account for potential lag effects between routine immunization performance and COVID-19 vaccination uptake.

Response: Our intention was to examine the link between the immunisation system performance immediately prior to the COVID-19 pandemic and COVID-19 vaccination coverage. We used 5-year mean annual estimates of coverage which would be a reasonable indication of the state of systems at the time of the pandemic.

Changes to manuscript: No changes made.

Comment 6: The study mentions supply issues. There are other important factors which are not considered in the current study including how geopolitical factors (e.g., vaccine diplomacy, reliance on external donors) impacted vaccine access in SIDS.

Response: We agree with the reviewer that these factors can play a role. However, for the purposes of this study, we wanted to examine the association between COVID-19 vaccination coverage and overall immunisation system performance. Examining all possible factors that may have influenced COVID-19 vaccination coverage was out of scope. Data on all these factors are also not consistently or systematically reported for SIDS. Noting the considerable shifts during the pandemic, these were difficult to measure with no baseline data.

We have noted the additional factors raised by the reviewer in the limitations section.

Changes to manuscript: Discussion section, page 31, lines 554–555, text added:

Other factors, such as policies related to mandatory vaccination requirements for work or trave

---

## [Decision Letter · Decision Letter 1]

Dear Dr. Patel,

Thank you for submitting your manuscript to PLOS ONE. After careful consideration, we feel that it has merit but does not fully meet PLOS ONE’s publication criteria as it currently stands. Therefore, we invite you to submit a revised version of the manuscript that addresses the points raised during the review process.

We look forward to receiving your revised manuscript.

Kind regards,

Harapan Harapan, MD, PhD

Academic Editor

PLOS ONE

Journal Requirements:

Reviewers' comments:

Reviewer's Responses to Questions

**Comments to the Author**

Reviewer #1: (No Response)

Reviewer #3: All comments have been addressed

Reviewer #4: (No Response)

Reviewer #5: (No Response)

Reviewer #6: All comments have been addressed

Reviewer #7: All comments have been addressed

2. Is the manuscript technically sound, and do the data support the conclusions?

Reviewer #1: Partly

Reviewer #3: Yes

Reviewer #4: Partly

Reviewer #5: Partly

Reviewer #6: Yes

Reviewer #7: Yes

3. Has the statistical analysis been performed appropriately and rigorously?

Reviewer #1: Yes

Reviewer #3: Yes

Reviewer #4: Yes

Reviewer #5: Yes

Reviewer #6: Yes

Reviewer #7: Yes

4. Have the authors made all data underlying the findings in their manuscript fully available?

Reviewer #1: Yes

Reviewer #3: Yes

Reviewer #4: Yes

Reviewer #5: Yes

Reviewer #6: Yes

Reviewer #7: Yes

5. Is the manuscript presented in an intelligible fashion and written in standard English?

Reviewer #1: Yes

Reviewer #3: Yes

Reviewer #4: Yes

Reviewer #5: Yes

Reviewer #6: Yes

Reviewer #7: Yes

Reviewer #1: 1. Table 1 does not add any extra information to your manuscript, the authors could add a brief of how the analysis made to the method section.

2. Could you give more details about the income levels in Table 2, how they have been classified to Low, Lower-middle, Upper-middle, and Other

3. Rewrite the conclusion section in your manuscript, indicates what is the main finding or conclusion from your study that summaries the result section

Reviewer #3: (No Response)

Reviewer #4: The ability of developing countries to adapt quickly and withstand changes is often limited by the lack of infrastructure, personnel, and adequate equipment in the health sector. I believe that this study is not only relevant, but essential, and should be approached from a critical-scientific perspective in order to promote substantial improvements and provide a solid foundation for future health policy recommendations and studies. Below, I present my comments with the aim of strengthening its structure and presentation:

Abstract

Introduction:

1.- It would be beneficial to include a more detailed characterization of the study population. Additionally, include a brief contextualization of the level of public trust in vaccines, public health policies, and the current state of the logistics system.

Discussion:

1.- Generally, in a scientific research, the discussion should be written in the third person to achieve a formal style and maintain objectivity.

2.- The paragraph:

"The finding that SIDS that did not experience declines in RI coverage during the pandemic achieved high COVID-19 vaccination coverage aligns with the principles of a “strong” health system being a resilient one. The absence of a decline in RI coverage observed in these countries does not imply that there were no disruptions to immunisation services during the pandemic, but more likely that they were able to reorganise and rapidly recover from any disruptions... "

Is too long and repetitive, with too many ideas competing for attention, which weakens the overall message.

3.- It would be possible to conduct multivariate analysis on a smaller sample with complete data, which could improve the analysis and address one of the study's limitations.

Conclusions:

1.-It would be beneficial to investigate whether SIDS cultural or context-specific factors within the sample may have influenced these results.

Reviewer #5: A review report of the manuscript entitled “The association between routine immunisation and COVID-19 vaccination in small island developing states”

- (lines 49-50) The current sentence fails to clearly explain the statistical tools used in the data analysis. Please revise it by mentioning on which program the analysis was done and the statistical power used to determine the significance.

- (lines 55-61) Please unify the statistical report written on each sentence. There’s still disparity in the current way of reporting, especially in the usage of “r”, which should be applied uniformly.

- (line 75) The introduction fails to include relevant literature. The authors should improve their literature review by including more relevant published papers. Additionally, besides summarizing previous work on the topic, the authors should link their approach with previous studies by highlighting the state-of-the-art and gap in the literature.

- Please give a bit of overview on what COVID-19 is in the introduction. For example, “COVID-19 is induced by a positive-sense single-stranded RNA virus known as severe acute respiratory syndrome coronavirus 2 (SARS-CoV-2). Individuals afflicted with COVID-19 may exhibit minor symptoms such as fever, cough, headache, and sputum production, which can escalate to more serious complications, including acute respiratory distress syndrome (ARDS) and mortality.” This info can be cited from a study by Duta et al. (2023) entitled “Essential oils for COVID-19 management: A systematic review of randomized controlled trials”.

- (line 143) What were the exclusion criteria of this study?

- (lines 160-163) How did the authors deal with inconsistently reported RI data measured through the WUENIC system?

- (line 188) It is unclear what kind of first vaccine dose the authors mean in this phrase: “The primary outcomes of interest were coverage of: A) the first dose…”

- Authors are suggested to proofread the manuscript after addressing all comments to avoid any typological, grammatical, and lingual mistakes and errors. For example, there’s an extra space on line 279 and a misspelling on line 280; repetition on lines 299 and 361.

- (lines 467-468) This sentence “Least developed countries, with weaker health systems,[54] had lower COVID-19 vaccination coverage throughout the study period.” will be more significant if it’s also supported by another relevant study. For example, a study by Rosiello et al. (2021) entitled “Acceptance of COVID-19 vaccination at different hypothetical efficacy and safety levels in ten countries in Asia, Africa, and South America”

Reviewer #6: Thank you for giving me the opportunity to review your manuscript and addressing the comments of the previous reviewers. I have no further questions or comments.

Reviewer #7: Author have answered all the discussion given before and provide a better explanation. Some added information are also helping to explain the main focus of this study.

**Do you want your identity to be public for this peer review?** For information about this choice, including consent withdrawal, please see our Privacy Policy

Reviewer #1: **Yes: ** Dr Omeed Darweesh

Reviewer #3: No

Reviewer #4: No

Reviewer #5: No

Reviewer #6: **Yes: ** Ibrahim Saleh

Reviewer #7: No

---

## [Author Response · Author response to Decision Letter 2]

14 May 2025

We thank the reviewers for the comments on our paper and for the opportunity to address them. We have clarified certain aspects of our paper and included additional details in the manuscript in response. Below, we provide responses to each of the reviewers’ comments and line number references where we have made changes, or point to where existing text in the manuscript addresses comments.

We note that there were no comments from reviewers #2 and #3.

Responses to reviewer 1

1. Table 1 does not add any extra information to your manuscript, the authors could add a brief of how the analysis made to the method section.

Thank you for this comment. Table 1 provides the details on the independent variables included in the study and information on data sources. Our manuscript includes a summary of this information in the methods in the sections describing independent variables (see lines 219-261) and the data sources (lines 179-201). However, we feel the detail in the table is necessary for clarity and transparency of our methods. Including all this information in text form would be tedious for readers. We have included the table so that our methods are sufficiently detailed for another researcher to replicate them. We have made some additional clarifications to the manuscript text on data sources.

Added text underlined (lines 179-201):

We extracted the following publicly available data (Table 1):

• Monthly COVID-19 vaccination coverage data for 2021 and 2022 from COVID-19 Vaccination Information Hub, collated from the WHO Joint Reporting Form (JRF) COVID-19 vaccination module.[30]

• WHO/UNICEF estimates of national immunisation coverage (WUENIC) data for the years 2015 to 2022, collated from countries annually via the JRF process. Data were sourced from the WHO Immunization Data Portal, including data on RI coverage, new vaccine introduction and immunisation system performance.[31] Data for RI coverage were not consistently reported for non-WHO member states, and were unavailable for new vaccine introduction status and immunisation system performance measures through the WUENIC system. Indicators on dropout of routine immunisation coverage and disruptions to routine immunisation coverage during the COVID-19 pandemic were calculated based on WUENIC estimates.

• World Bank Indicators data for health system performance (e.g. health worker density and childhood mortality rates) and country-level demographic characteristics (e.g. country population size).

• Country income level and development status classifications as defined by the World Bank and United Nations Development Programme, respectively.

Table 1 provides further details on data sources for independent variables. All data were extracted in April 2024. As all data were aggregated at the country-level and obtained from publicly available sources, ethical approval was not required.

2. Could you give more details about the income levels in Table 2, how they have been classified to Low, Lower-middle, Upper-middle, and Other

Country income level categories are based on the World Bank’s classifications using the World Bank Atlas Method, which uses Gross National Income per capita to categories countries. Their methodology is detailed in this link: https://datahelpdesk.worldbank.org/knowledgebase/articles/906519-world-bank-country-and-lending-groups

We have clarified this in our manuscript text (line 196-197) and the footnote to Table 1 (line 205-206).

Edited text underlined:

Line 196-197: Country income level and development status classifications as defined by the World Bank and United Nations Development Programme, respectively.

Line 212-213: Country income level categories are defined by Gross National Income (GNI) per capita based on the World Bank’s classifications using the World Bank Atlas Method.

3. Rewrite the conclusion section in your manuscript, indicates what is the main finding or conclusion from your study that summaries the result section

We have edited the conclusion section to summarise the main findings and conclusions from the study.

Added text, line 581-583: In conclusion, we found that countries that achieved higher COVID-19 vaccination coverage also sustained pre-pandemic RI coverage during the pandemic, had introduced newer vaccines into their national immunisation schedules, and had higher workforce density.

Responses to reviewer 4

The ability of developing countries to adapt quickly and withstand changes is often limited by the lack of infrastructure, personnel, and adequate equipment in the health sector. I believe that this study is not only relevant, but essential, and should be approached from a critical-scientific perspective in order to promote substantial improvements and provide a solid foundation for future health policy recommendations and studies. Below, I present my comments with the aim of strengthening its structure and presentation:

Thank you for this feedback.

Abstract

Introduction:

1.- It would be beneficial to include a more detailed characterization of the study population. Additionally, include a brief contextualization of the level of public trust in vaccines, public health policies, and the current state of the logistics system.

We have added some additional details as requested.

Additional text added:

Lines 113-116: SIDS vary in their income level classifications from low through to high income, but are all considered to be vulnerable to systemic shocks due to social, economic, environmental and infrastructural challenges and limited resources.[9]

Lines 132-135: Additionally, they face higher freight costs to transport vaccines due to their remote geography. Vaccine distribution to the point-of-service is complicated by the need to maintain cold chain in warm climates and transporting vaccines by air and boat especially to remote outer lying islands.

Lines 139-150: Policies regarding border closures, quarantine protocols and vaccine mandates varied, as did decision-making and emergency response coordination structures.[15,20] For example in the Pacific, country-level responses were complemented by the WHO’s wider Pacific Joint Incident Management Team partner coordination meetings.[20] In the Caribbean, the Caribbean Public Health Agency Caribbean Regulatory System collaborated with the WHO and Pan American Health Organization to expedite regulatory approvals of COVID-19 vaccines.[23] Acceptance of COVID-19 vaccination was also a challenge in some SIDS, with surveys in Papua New Guinea, Haiti and Dominican Republic indicating only 27.6%, 43.2% and 68.8% of individuals, respectively, intended to receive a COVID-19 vaccine.[24,25] Another survey of healthcare workers found broad (>90%) acceptance of vaccines in general, but hesitancy to receive COVID-19 vaccines among 23% of respondents.[26]

Discussion:

1.- Generally, in a scientific research, the discussion should be written in the third person to achieve a formal style and maintain objectivity.

Thank you for this feedback. We have reviewed the manuscript and believe that the language used in the discussion is appropriate. The discussion is written in a formal academic voice, and we have maintained consistency throughout. The style and tone are similar to other recently published articles in PLoS One, e.g.:

• Brehon et al. PLoS One, 2025; 20(5): e0322911 https://doi.org/10.1371/journal.pone.0322911

• Hiranburana et al. PLoS One, 2025; 20(4): e0317940. https://doi.org/10.1371/journal.pone.0317940

• Acharya et al. PLoS One, 2025; 20(4): e0322031. https://doi.org/10.1371/journal.pone.0322031

We appreciate preferences for writing style vary and will work with the editorial team to finalise the manuscript.

2.- The paragraph:

"The finding that SIDS that did not experience declines in RI coverage during the pandemic achieved high COVID-19 vaccination coverage aligns with the principles of a “strong” health system being a resilient one. The absence of a decline in RI coverage observed in these countries does not imply that there were no disruptions to immunisation services during the pandemic, but more likely that they were able to reorganise and rapidly recover from any disruptions... "

Is too long and repetitive, with too many ideas competing for attention, which weakens the overall message.

We have revised this paragraph and tried to simplify the message.

Revised text underlined, lines 464-488:

We found that SIDS that did not experience declines in RI coverage during the pandemic also achieved high COVID-19 vaccination coverage. This supports the principle that a “strong” health system is a resilient one. The absence of a decline in RI coverage observed in these countries does not imply that there were no disruptions to immunisation services during the pandemic. Rather, it is more likely that they were able to reorganise and rapidly recover from any disruptions. This ability to rapidly recover from shocks to the system while responding to the crisis is a defining characteristic of resilient health systems.[46]

Resilience is dependent on having a high functioning and adaptive health system, underpinned by equitable access to essential health services and a skilled and adequately sized workforce.[46] While higher COVID-19 vaccination coverage was correlated with the workforce density and having introduced and sustained newer vaccines, we did not find an association between universal health coverage and COVID-19 vaccination coverage. This is contrary to a recent study that demonstrated countries with higher universal health coverage index scores had higher rates of COVID-19 vaccination.[47] Our findings might reflect intensified international development partner efforts in response to COVID-19 especially in lower income countries, or the public response to COVID-19 control measures, such as requirements for vaccination certificates for travel.[48]

3.- It would be possible to conduct multivariate analysis on a smaller sample with complete data, which could improve the analysis and address one of the study's limitations.

A rule of thumb in multivariate regression is that there should be at least 10 observations for every independent variable included in the model. The degree of missing data meant that any multivariate analysis we conducted could not have included more than 3 independent variables. For example, 13 of 55 SIDS (23.6%) had not reported any data on BCG vaccination coverage for the 5-year period from 2015 to 2019. Fifteen (27.3%) did not report any data on hepatitis B birth dose vaccination coverage for the same period. Data on new vaccine introductions was only available for SIDS who were WHO member states (39/55, 70.9%). As we would have needed to control for country-level statistics like income level and country size, the small sample size meant we could not to explore the relationship between RI variables in any further depth.

Conclusions:

1.-It would be beneficial to investigate whether SIDS cultural or context-specific factors within the sample may have influenced these results.

We agree these could have affected COVID-19 vaccination uptake. However, this study primarily examined the association between COVID-19 vaccination coverage and overall immunisation system performance. Examining all possible factors that may have influenced COVID-19 vaccination coverage was out of scope. Data on all these factors are also not consistently or systematically reported for SIDS. We have noted this as a limitation in our discussion section, please see lines 568-577 copied below.

Existing text: Additional factors that could be examined include service delivery and accessibility, vaccine acceptance and demand, and cold chain capacity, which were beyond the scope of this study. Cultural and political factors also have a role, with multiple studies reporting that trust in government was strongly associated with higher, faster and earlier uptake of COVID-19 vaccination and other outcomes during the pandemic including COVID-19 mortality.[8,67,68] Other factors, such as policies related to mandatory vaccination requirements for work or travel, geopolitical factors including vaccine diplomacy, and the role and influence of external donors and global health partner organisations could have also influenced COVID-19 vaccine uptake.

Responses to reviewer 5

A review report of the manuscript entitled “The association between routine immunisation and COVID-19 vaccination in small island developing states”

- (lines 49-50) The current sentence fails to clearly explain the statistical tools used in the data analysis. Please revise it by mentioning on which program the analysis was done and the statistical power used to determine the significance.

We have edited the sentence to clarify our statistical approach.

Edited text underlined, line 49-50: We calculated Spearman correlation coefficients (r) with p-values (p<0.05 considered significant) and 95% confidence intervals for continuous variables and mean COVID-19 vaccination coverage by categorical variables.

- (lines 55-61) Please unify the statistical report written on each sentence. There’s still disparity in the current way of reporting, especially in the usage of “r”, which should be applied uniformly.

We have edited the abstract to clarify our statistical approach, specifically our use of correlation coefficients. In the results section of the abstract, we have checked that we have consistently reported the r value and p-value for the relevant statistics.

Edited text underlined, line 49-50: We calculated Spearman correlation coefficients (r) with p-values (p<0.05 considered significant) and 95% confidence intervals for continuous variables and mean COVID-19 vaccination coverage by categorical variables.

- (line 75) The introduction fails to include relevant literature. The authors should improve their literature review by including more relevant published papers. Additionally, besides summarizing previous work on the topic, the authors should link their approach with previous studies by highlighting the state-of-the-art and gap in the literature.

Our paper examines the relationship between routine immunisation systems and emergency vaccination, specifically COVID-19 vaccination. Only one other study has quantitatively examined this relationship in a multi-country analysis to date, which we have summarised in lines 96-102. We have reviewed our introduction section to ensure the relevant existing literature is cited.

- Please give a bit of overview on what COVID-19 is in the introduction. For example, “COVID-19 is induced by a positive-sense single-stranded RNA virus known as severe acute respiratory syndrome coronavirus 2 (SARS-CoV-2). Individuals afflicted with COVID-19 may exhibit minor symptoms such as fever, cough, headache, and sputum production, which can escalate to more serious complications, including acute respiratory distress syndrome (ARDS) and mortality.” This info can be cited from a study by Duta et al. (2023) entitled “Essential oils for COVID-19 management: A systematic review of randomized controlled trials”.

We have added an overview of COVID-19.

Added underlined text, line 91-95: The COVID-19 pandemic provides an opportunity to apply a health systems lens to examine the link between RI performance and emergency vaccination. COVID-19, caused by the severe acute respiratory syndrome coronavirus 2 (SARS-CoV-2), can cause severe respiratory disease requiring hospitalisation and critical care such as ventilation, and can be fatal.[7] Its high transmissibility threatened to quickly overwhelmed health systems in the first wave of infection, leading to intense efforts to develop and deploy an effective vaccine. A recent analysis of COVID-19 vaccination coverage and immunisation program maturity found that COVID-19 vaccination coverage was 14-16% higher in countries with an adult seasonal influenza vaccination program.

- (line 143) What were the exclusion criteria of this study?

We have added a statement to make our exclusion criteria clear.

Added text (line 176-177): We only excluded SIDS for which COVID-19 vaccination coverage was not publicly reported.

- (lines 160-163

---

## [Decision Letter · Decision Letter 2]

The association between routine immunisation and COVID-19 vaccination in small island developing states

PONE-D-24-59520R2

Dear Dr. Patel,

We’re pleased to inform you that your manuscript has been judged scientifically suitable for publication and will be formally accepted for publication once it meets all outstanding technical requirements.

Kind regards,

Harapan Harapan, MD, PhD

Academic Editor

PLOS ONE

Additional Editor Comments (optional):

Reviewers' comments:

Reviewer's Responses to Questions

**Comments to the Author**

Reviewer #3: All comments have been addressed

Reviewer #4: All comments have been addressed

Reviewer #6: All comments have been addressed

2. Is the manuscript technically sound, and do the data support the conclusions?

Reviewer #3: (No Response)

Reviewer #4: Yes

Reviewer #6: Yes

3. Has the statistical analysis been performed appropriately and rigorously?

Reviewer #3: (No Response)

Reviewer #4: Yes

Reviewer #6: Yes

4. Have the authors made all data underlying the findings in their manuscript fully available?

Reviewer #3: (No Response)

Reviewer #4: Yes

Reviewer #6: Yes

5. Is the manuscript presented in an intelligible fashion and written in standard English?

Reviewer #3: (No Response)

Reviewer #4: Yes

Reviewer #6: Yes

Reviewer #3: (No Response)

Reviewer #4: All reviewer comments have been addressed in a timely and thorough manner. In my assessment, the manuscript is now suitable for publication. I would like to express my gratitude to the editor for the opportunity to contribute as a reviewer.

Reviewer #6: Thank you for giving me the opportunity to review your manuscript entitled "The association between routine immunisation and COVID-19 vaccination in small island developing states," and addressing the

comments of the previous reviewers. I have no further questions or comments.

**Do you want your identity to be public for this peer review?** For information about this choice, including consent withdrawal, please see our Privacy Policy

Reviewer #3: No

Reviewer #4: **Yes: ** Araceli Guerra Martínez

Reviewer #6: **Yes: ** Ibrahim Saleh

---

## [Editor Report · Acceptance letter]

PONE-D-24-59520R2

PLOS ONE

Dear Dr. Patel,

I'm pleased to inform you that your manuscript has been deemed suitable for publication in PLOS ONE. Congratulations! Your manuscript is now being handed over to our production team.

Kind regards,

on behalf of

Dr. Harapan Harapan

Academic Editor

PLOS ONE